# NeuroExaminer: an all-glass microfluidic device for whole-brain in vivo imaging in zebrafish

Kai Mattern[1,3], Jakob William von Trotha [2,3], Peer Erfle [1], Reinhard Wolfgang Köster [2] & Andreas Dietzel [1✉]

While microfluidics enables chemical stimuli application with high spatio-temporal precision, light-sheet microscopy allows rapid imaging of entire zebrafish brains with cellular resolution. Both techniques, however, have not been combined to monitor whole-brain neural activity yet. Unlike conventional microfluidics, we report here an all-glass device (NeuroExaminer) that is compatible with whole-brain in vivo imaging using light-sheet microscopy and can thus provide insights into brain function in health and disease.

[1] Institute of Microtechnology (IMT), Technische Universität Braunschweig, Braunschweig, Germany. [2] Division of Cellular and Molecular Neurobiology, Zoological Institute, Technische Universität Braunschweig, Braunschweig, Germany. [3] These authors contributed equally: Kai Mattern, Jakob William von Trotha. ✉email: a.dietzel@tu-braunschweig.de

The zebrafish was the first vertebrate organism in which the neural activity of nearly the entire brain could be observed in vivo at single cell resolution approximately every second[1]. More recently, microfluidics has discovered this small vertebrate as a suitable model system to investigate brain function and to screen for new drugs to alleviate psychiatric diseases[2–6]. The ability of microfluidics to deliver chemical stimuli with millisecond precision, together with the zebrafish's optical translucency during its larval stages thus provide an outstanding opportunity to investigate the dynamics and interactions of neural circuits in a non-invasive manner in an intact living organism. Here, we report an all-glass microfluidic device that, in contrast to the more commonly used polydimethylsiloxane (PDMS), does not exhibit solvent permeability[7] or autofluorescence[8], and is therefore compatible with whole-brain in vivo imaging via light-sheet microscopy.

## Results

**Microfluidic chip design and simulations.** For computational fluid dynamics simulations (CFD software 18.0 from Ansys, Canonsburg, USA) three-dimensional models of the microfluidic system with the enclosed larva were created using computer-aided design (CAD software from SolidWorks, Vélizy-Villacoublay, France). It is worth noting that horizontal mounting of the larva represents a more physiological orientation compared to the vertical mounting approach used in early whole brain-imaging studies[1]. The larva and the glass walls were assumed as non-deforming solid bodies. Two fluid phases exhibiting the characteristic properties of water at a temperature of 28 °C were considered in the simulations as representing media and chemical stimuli. Transient state simulations of the injection of chemical stimuli over a period of 1.5 s (at 50 ms steps) were performed (Fig. 1a–f). The simulations were carried out for advancing design versions, whereby the position of the bypass opening as well as the size and positioning of stimulus and media injection channels were progressively improved to enable a fast exchange of fluids and three-dimensional flow focusing of the stimulus phase. The media inlet divides into an upper and a lower channel (Fig. 1g), which focuses the stimulus injection positioned in between. The lower media inlet channel exhibits a slightly higher hydrodynamic resistance, so that the injected stimulus is initially directed slightly downwards within the alignment chamber. The flow of medium introduced at the bottom of the alignment chamber helps to reduce the loss of stimulus that would otherwise be conducted directly out of the system via the bypass. Both of these features ensure that the stimulus reaches the larvae in a targeted manner. CFD simulations of two-system variants (Fig. 1a–f; Supplementary Movies 1 and 2) support the targeted exposure of the larva head to the stimulus under conditions of continuous supply of fresh media. The partially open system variant (Fig. 1c, d, f) exhibits a lower hydrodynamic resistance due to a recess area above the head of the larva. This additional opening deflects the current of the stimulus in an upward direction. With the additional opening the counter pressure that retains the larva is reduced which may be compensated by a higher throughput of stimuli and medium. In the simulations, both NeuroExaminer variants enable a precise spatial and temporal control (with subsecond resolution) of exposure to varying stimuli. With the opening though, the flow in the bypass channel practically stops and parts of the larva are less exposed to fresh media fluid. The partially open variant is advantageous for optical imaging, since no microfabricated glass material is introduced in the imaging light path, while the completely closed variant (Fig. 1a, b, e) enables unparalleled control over stimuli exposure of the larva and is ideally suited for precise time-resolved

compound stimulation. The simulations shown in Fig. 1a–f represent a total volume flow rate (with media and stimulus injection at a ten-to-one ratio) of 1.1 µl/s. The Reynolds numbers, which indicate the balance between viscous forces and inertial forces, were confirmed to stay below $Re = 10$. Hence, a completely laminar flow pattern without backflows or vortices is observed that allows 3D focusing of the stimulus for precise targeting of the chemical stimuli. Transient simulations revealed that a 100% concentration of stimulus reaches the larva head in <1 s after the injection into the alignment chamber in both the open and closed system (Fig. 1a–d; Supplementary Movies 1 and 2). Based on these simulations, which can very reliably predict the behavior in laminar flow experiments, it can be concluded that the desired precise spatial and temporal control of the stimulus delivery in the NeuroExaminer is indeed possible. The complex 3D chamber was designed to gently immobilize zebrafish larvae for high-resolution imaging (Fig. 1h illustrates the incoupling of the light sheet and the positioning of the larva under the detection objective) and to subsequently release them without harm (see also Supplementary Fig. 1). The CFD simulations support the chosen 3D design including microchannels used as inlets, outlets, and for stimulus application as shown in Fig. 1g. The 3D chamber design follows the concept of reversible fluidic larva immobilization (Fig. 1i–l) enabling: (Fig. 1i) zebrafish larvae loading through fluidic inlets, (Fig. 1j) flow-mediated autonomous animal alignment, (Fig. 1k) reversible larva fixation through reversed flow from the outlet, and (Fig. 1l) controlled exposure to stimuli. Small stimuli volume flows of 0.1 µl/s or less in combination with higher media volume flows of 1 µl/s or more not only considerably reduce the consumption of potentially expensive substances, but also facilitate rapid fluid exchange under laminar conditions. The microfluidic chip (NeuroExaminer) is part of a larger ensemble and embedded in a custom-made imaging chamber; Supplementary Fig. 1 provides an overview of the setup used in this study.

Next, we constructed the two variants of NeuroExaminer chips solely made from glass using femtosecond laser processing and subsequent thermal and chemical surface quality enhancement. Multi-photon-absorption in glass allows iterative ablation used for creating three-dimensional microfluidic structures (see "Methods" section for details). After structuring, the two system halves are bonded to obtain closed microfluidic chips (Fig. 1m, n).

**Light sheet imaging in the microfluidic chip.** To test the imaging properties of the microfluidic devices, 6 days post fertilization (dpf) zebrafish larvae expressing the genetically encoded $Ca^{2+}$-indicator GCaMP6s under the control of the pan-neuronal elavl3 (formerly HuC) promoter[9] in the crystal background[10] were injected into the NeuroExaminer and oriented manually (Fig. 2a, e; Supplementary Movie 8). After an experiment the larva could easily be flushed out of the device to image another larva. For whole-brain imaging, we collected 21 optical sections every 10 µm at nearly 0.3 Hz via digital light sheet (DLS) microscopy for more than 10 min (see "Methods" section for details). We monitored and compared neuronal activity and image quality in both an open ($n = 11$ fish; Fig. 2b–d; Supplementary Movies 3 and 4) and a closed system ($n = 15$ fish; Fig. 2f–h; Supplementary Movies 5 and 6). While the open system contains a recess area just above the larva's head, in the closed system this area is enclosed by borosilicate glass (compare Fig. 2a, e). In both systems, the resulting images enabled single cell resolution of $Ca^{2+}$-transients at about 0.3 Hz (Supplementary Movie 7). Faster volumetric imaging at about 1 Hz, however, is limited to about five optical planes on the Leica SP8 DLS (Supplementary Fig. 2 and Supplementary Movies 9 and 10). In the open system, single-cell

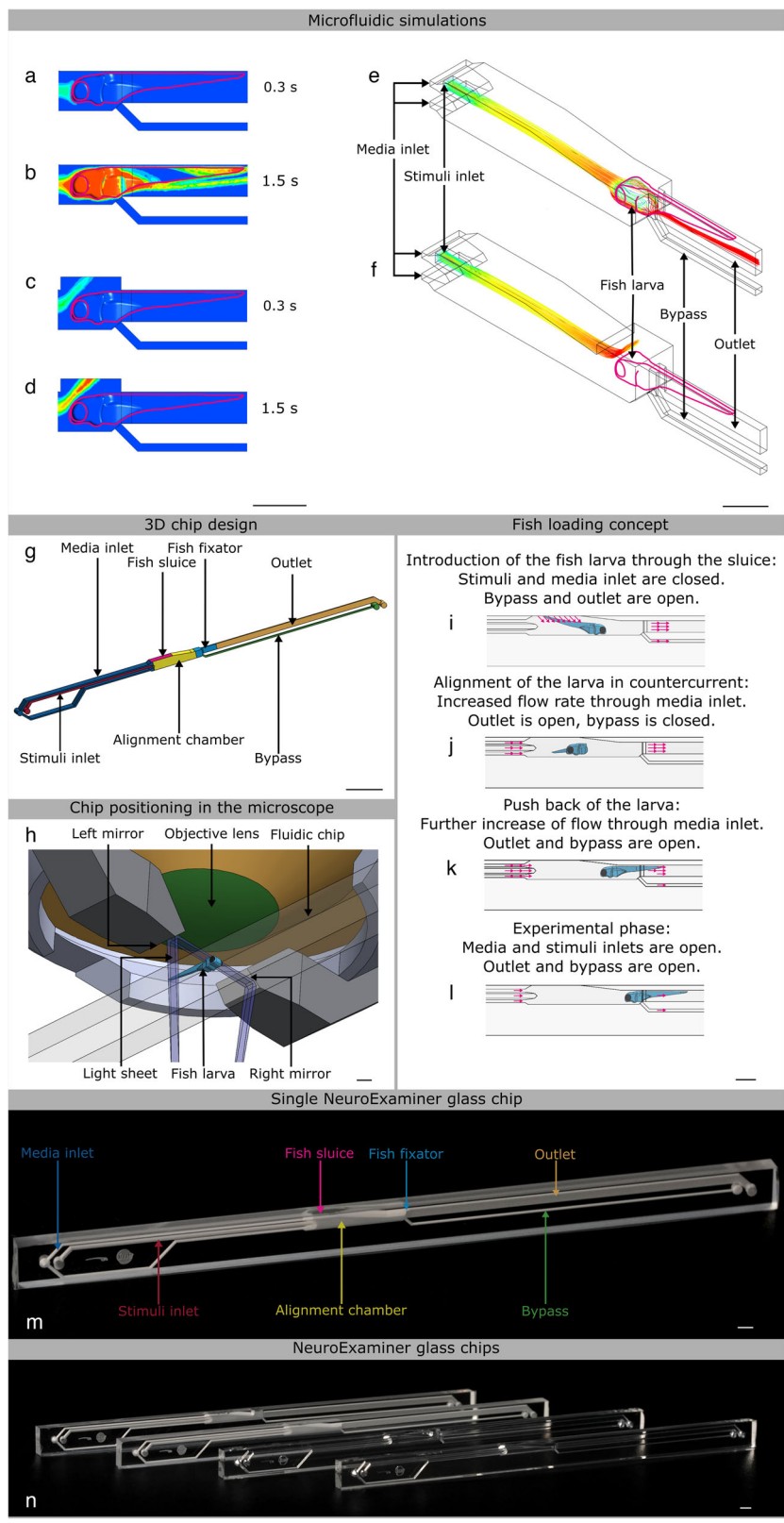

resolution could be obtained throughout all layers (Fig. 2c, d) enabling whole-brain 4D reconstruction of $Ca^{2+}$-signals at a temporal resolution of 1 brain volume nearly every 3 s (Supplementary Movies 3 and 4). The closed system provided this resolution for volume layers of about half of the brain (100 μm) (Fig. 2g, h) with $Ca^{2+}$-signals emanating from further ventral structures still being detectable but appearing weaker and spatially less well resolved (compare z-planes at 120 and 180 μm in Fig. 2c, d with Fig. 2g and h). Clearly, individual neurons throughout the brain could be allocated to distinct brain regions and neural clusters and their resting activity could be resolved (Fig. 2i–m; Supplementary Fig. 2; Supplementary Movies 7 and 10).

**Fig. 1 NeuroExaminer design and fabrication. a–d** Simulation of the targeted stimulus injection at a volume flow rate of 0.1 µl/s with simultaneous supply of oxygen-enriched medium at 1 µl/s in a cross-sectional view in the closed **a**, **b** and open **c**, **d** NeuroExaminer versions at 0.3 and 1.5 s after the entry of the stimulus into the alignment chamber. The color-coded distribution shows the concentration (blue = 100% medium and red = 100% stimulus) in the mid-plane of the channel, and on the larval surface after impacting the larva. **e**, **f** 3D view of stimulus streamline patterns of injected stimuli (blue = 0 mm/s, red = 25 mm/s) in the closed **e** and open **f** system variant simulated assuming the same inflow volume currents as in **a–d**. The larva head experiences a homogeneous stimulus exposure in the closed system, whereas in the open system the stimulus is also directed outside of the device reducing the contact surface of the larva head with the stimulus solution. **g** 3D design of the closed system variant. **h** Schematic 3D view of the zebrafish larva positioned inside the fluidic chip and the light sheet microscope setup (see also Supplementary Fig. 1). **i–l** Illustration of the four-step loading concept. **m** A photo of the closed NeuroExaminer chip variant taken after thermal bonding. **n** The influence of the post treatment process on optical transparency illustrated by a photo comparing NeuroExaminer chips representing (from left to right) the open system before and after the combined chemical and thermal treatments and the closed system before and after the treatment. The scale bar is 1 mm throughout, except for **g** where it is 10 mm.

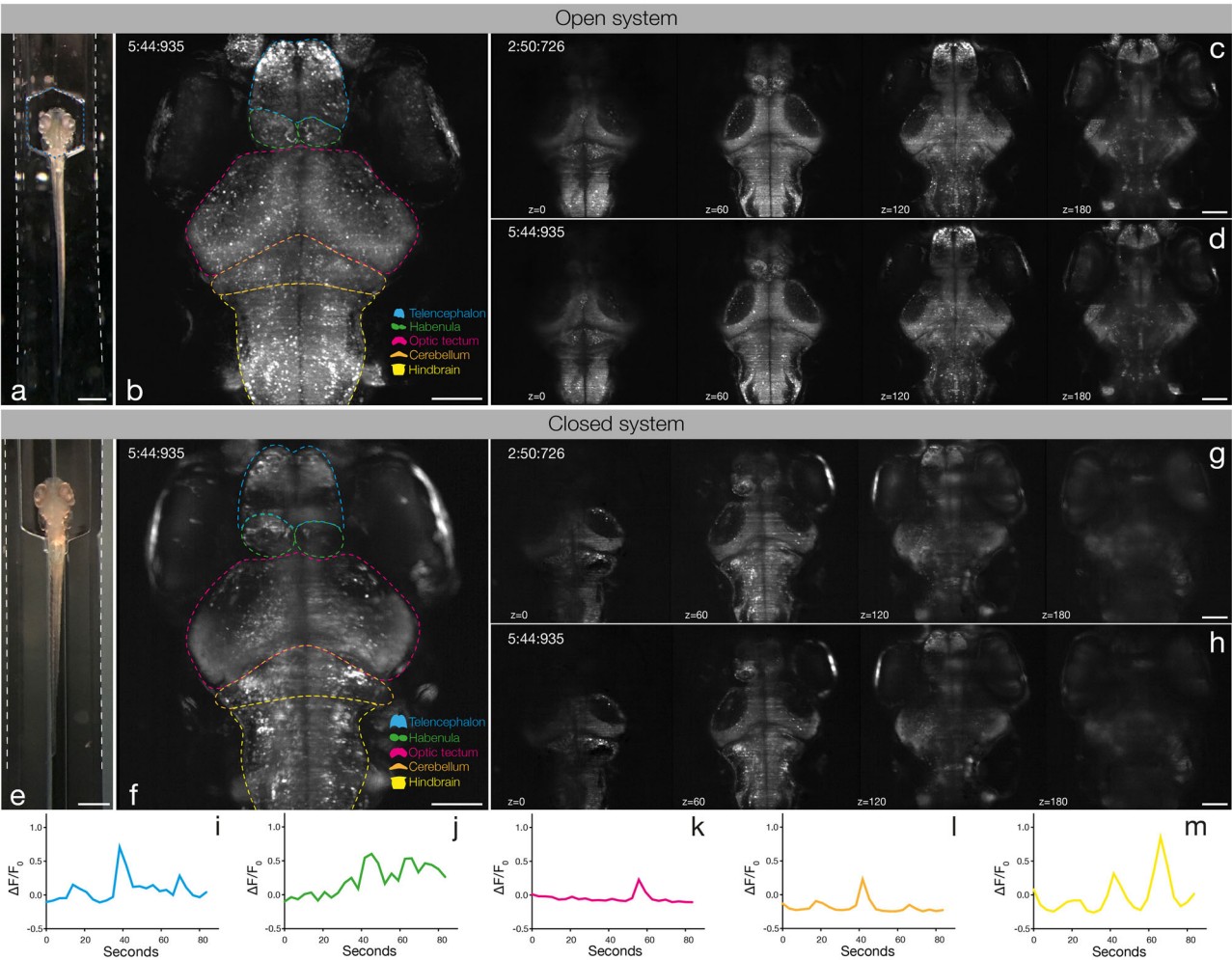

**Fig. 2 Whole-brain in vivo imaging in the NeuroExaminer. a, e** A live 6 dpf Tg(elavl3:H2B-GCaMP6s); crystal larva in an open **a** or closed **e** version of the NeuroExaminer. The outer borders of the open **a** or closed **e** microfluidic device are outlined with dotted white lines; the recess area above the larva's head in **a** is marked by a dotted blue line (see also Supplementary Movie 8). **b, f** A maximum intensity projection consisting of 21 optical sections at 5 min and 45 s (time is indicated in min:s:ms) depicting nuclear-localized GCaMP6s throughout the larva's brain in the open **b** and closed **f** system (see also Supplementary Movies 3, 4). Dotted color-coded lines outline five major brain regions depicted on the lower right. **c, d** and **g, h** Neural activity patterns at four single optical sections spaced 60 µm apart at two indicated time points (shown in min:s:ms) in the open **c, d** and closed **g, h** NeuroExaminer (see also Supplementary Movies 5, 6). Note that optical sections in the open system are spatially well-resolved throughout the entire brain **c, d**, whereas in the closed system dorsal sections display a spatially better resolution than ventral sections localized deeper in the brain **g, h**. **i–m** Single-cell color-coded calcium signals (25 frames; ~84 s) for the fish in the closed system in the telencephalon (**i**, blue), habenula (**j**, green), optic tectum (**k**, magenta), cerebellum (**l**, orange), and hind brain (**m**, yellow; see also Supplementary Movie 7). Scale bar is 500 µm in **a** and **e** and 100 µm in **b–d** and **f–h**.

## Discussion

Future versions of the NeuroExaminer will be used to investigate the response of the whole brain and the interaction of neural circuits within it to specific stimuli. Although we have tested our microfluidic chip thus far only with a single commercial light sheet microscope (Leica SP8 DLS), it is worth noting that it should be compatible with a variety of different commercial and non-commercial light sheet setups that allow for even faster

volumetric imaging (several brain volumes per second). Moreover, with the gentle fluidic manipulation and fixation of zebrafish larvae in a natural upright position that does not require embedding in agarose, single larvae can even be reused at a later time for consecutive long-term monitoring. This may be especially useful to see how neural circuits are modified or adapt to a repeated application of neuromodulatory compounds over time. Furthermore, the NeuroExaminer can thus be equipped with automation for larva loading and ejection that will allow for moderate compound throughput in whole-brain physiological analysis at cellular resolution in the scale of seconds. High-throughput sequential imaging of several larva with an upscaled device will be of significant interest to screening studies. Together, our microfluidic chip will provide a unique and strong tool for neuromodulatory compound characterization, connectome analysis, as well as drug validation and quality control.

## Methods

**Microdevice fabrication**. The structuring of glass (700 μm-thick BOROFLOAT® substrates from Schott AG, Mainz, Germany) was carried out in a laser workstation (Microstruct-C from 3D-Micromac, Germany) equipped with femtosecond YB:KGW laser (Pharos from Light Conversion, Vilnius, Lithuania) operated at the fundamental wavelength of 1030 nm. The beam was scanned over the substrate surface at 2000 m/s focused with an F-theta lens of 100 mm focal length. The 3D-design was converted into multiple layers of 50 μm height for laser processing. The areas to be removed were filled with scan lines with a distance of 4 μm starting at a distance of 4 μm from the desired contour edge. Each of these sets was rotated 30° against the previous one. The laser was operated at a repetition rate of 600 kHz and emitted pulses of 215 fs with an energy of 14.65 μJ. After ablation, the system halves were cleaned in an ultrasonic bath filled with ethanol for 15 min, then immersed for 0.5 min in a solution of 45 ml $H_2O$, 100 ml $H_3PO_4$, and 30 ml hydrofluoric acid and finally subjected to cleaning in a spray processor unit (Fairchild Convac, Neuenstadt, Germany) using distilled water and a mixture of $H_2SO_4$ and $H_2O_2$. After alignment of both system halves (in mask aligner EVG 620 from EV Group, St. Florian am Inn, Austria) a pre-bonding force was applied manually. The final thermal bonding was carried out at 630 °C for 6 h with a pre-heating phase at 600 °C for 15 min while a force of 4 kN was uniformly applied to the entire 4″ wafer surface. After separation with a wafer saw (DAD320, Disco Corporation, Tokyo, Japan), the individual chips (exemplary chip shown in Fig. 1m) were rinsed with water. Chips were dehydrated on a hotplate at 120 °C for 5 min before they underwent a heat treatment at 740 °C for 1 h in the muffle furnace. This heat treatment was performed twice to establish smooth glass surfaces in the microchannels (Fig. 1n illustrates the improvement in optical transparency). Finally, the chips were cleaned in an ultrasonic bath.

**Zebrafish maintenance**. Zebrafish (*Danio rerio*) were maintained at 28.5 °C on a 14-h light/10 h dark cycle and bred following standard procedures[11]. Transgenic zebrafish larvae *Tg(elavl3:H2B-GCaMP6s)*[9] in the *crystal* (*alb*b4/b4; *nacre*w2/w2; *roy*a9/a9) background[10] were raised in modified Danieau solution (29 mM NaCl, 0.7 mM KCl, 0.6 mM Ca(NO$_3$)$_2$, 0.4 mM MgSO$_4$, 5.0 mM HEPES, pH 7.0). All animal procedures and experiments were conducted in accordance with the European Union Directive 2010/63/EU to reduce and minimize animal suffering and were approved by the Lower Saxony State Office for Consumer Protection and Food Safety (33.19-42502-04-17/2533).

**Imaging acquisition and data processing**. Open and closed microfluidic devices (NeuroExaminer) were placed into a custom-made aluminum imaging chamber filled with ~8 ml of Danieau solution. 6 dpf *Tg(elavl3:H2B-GCaMP6s)*; *crystal* larvae were briefly (~1 min) anaesthetized in a volume of 100 μl Danieau solution containing 0.5 mg/ml Mivacurium chloride[12] (Santa Cruz Biotechnology, Dallas, TX, USA; sc-204809A). A single anaesthetized larva was injected via pipetting into the NeuroExaminer, carefully oriented in the adjustment chamber, and placed into the fish fixator with its tail pointing towards the outlet with a small Microloader$^{TM}$ plastic pipetting tip (Eppendorf, Wesseling-Berzdorf, Germany, #5242956003). Imaging was performed on a Leica TCS SP8 digital light sheet (DLS) microscope (Leica microsystems, Wetzlar, Germany) using a ×2.5 illumination (NA 0.07) and a ×10 detection objective (NA 0.3). The exposure time was set to 19.85 ms, with a light sheet thickness of 10 μm ("extended"); 2 × 2 binning produced an image with a voxel size of 0.719 μm for x and y dimensions. A total of 21 optical sections each 10 μm apart were recorded from both mirrors (xymzt mode in the LAS X software) resulting in a volume of 735 × 735 × 200 μm³. This whole-brain volume was recorded every 3.48 s (0.29 Hz) for a duration of 200 frames corresponding to 11 min and 35 s. Following image recording, signals from the left and right mirrors were merged using the "Fusion-Wavelet" function of the LAS X software to improve image quality and reduce file size. Leica .xlef files were converted in .lif files and the latter were loaded into Imaris 9.1.2 (Bitplane, Zurich, Switzerland). Imaris was used to visualize and process the 4D datasets and to create movies (see Supplementary Movies 3–7 and 9, 10) together with Fiji (https://fiji.sc/), Adobe Photoshop CC (19.1.8), and Quicktime 7.6.3 Pro (Apple, Cupertino, CA, USA). Bright field images (Fig. 2a, e and Supplementary Movie 8) were recorded with an iPhone SE (first generation) attached to LabCam Pro (iDu Optics, Detroit, MI, USA) on a Leica MZ 12.5 or Leica MZ 6 stereomicroscope, or Leica SP8 DLS. Images were processed and assembled using Adobe Photoshop CC (19.1.8) and Illustrator CC (22.1), respectively. Pictures of the imaging setup (Supplementary Fig. 1b–d) were acquired with a Nikon D7100 digital single lens reflex camera (DSLR) and a 40 mm 1:2.8 G micro-Nikkor objective.

**Calcium signal analysis**. Calcium signals in the closed NeuroExaminer were extracted from individual neurons on a single optical section (z-plane 8 (70 μm)) and analyzed in five major brain areas (telencephalon; habenula; optic tectum; cerebellum; hind brain) using Fiji. Regions of interest (ROI) were manually drawn over clearly identifiable neurons using the oval selection tool and their mean gray values were measured between frame 21 and 50 (~101 s). The values from the first five frames (21–25) were averaged and defined as $F0$, and calcium activity was then calculated for the frames 26–50 as $(F−F0)/F0 = \Delta F/F0$ (Fig. 2i–m; Supplementary Data 1). To extract calcium traces from single neurons recorded at 1 Hz (Supplementary Fig. 2; Supplementary Movies 9, 10) we restricted imaging to five optical planes with an exposure time of 17 ms. ROIs were manually drawn and analyzed on a single optical (z plane 2 (10 μm)) as described above for 115 frames (~103 s; Supplementary Data 2). Graphs for calcium traces were created with Prism 8.2.1 (GraphPad, La Jolla, CA, USA) and Adobe Illustrator CC (22.1).

**Statistics and reproducibility**. Two to four 6 dpf *elavl3:H2B-GCaMP6s* larvae in the *crystal* background were imaged between 10 a.m. and 8 p.m. on a single day in the final versions of the closed or the open microfluidic chip. The imaging was repeated on several days. A total of 11 and 15 larvae were imaged in the open and closed microfluidic chip, respectively.

**Reporting summary**. Further information on research design is available in the Nature Research Reporting Summary linked to this article.

## Data availability

All relevant data generated or analyzed during this study are included in this published article and its supplementary information files.

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

## Acknowledgements

We are grateful to Misha Ahrens for the *Tg*(*elavl3:H2B-GCaMP6s*) line and to Paride Antinucci for providing the *crystal* fish. We thank Jomo Walla and Manuel Hohgardt for their help to determine the point spread function (PSF), and Timo Fritsch for excellent animal care. The work was funded in part by the DFG—Deutsche Forschungsgemeinschaft (KO1949/7-2, Project No. 241961032). K.M. was financed through the EXIST program (Grant No. 031L0149) supported by the Deutsches Bundesministerium für Wirtschaft und Energie. P.E. was financed by QUANOMET-(Programm der Niedersächsischen Wissenschaftsallianz, Grant No. ZN3378) provided by the Ministerium für Wissenschaft und Kultur des Bundeslandes Niedersachsen. We acknowledge support by the German Research Foundation and the Open Access Publication Funds of the Technische Universität Braunschweig.

## Author contributions

K.M., J.W.v.T., R.W.K., and A.D. conceived the basic idea of this interdisciplinary project and continuously refined the NeuroExaminer concept together. K.M. performed the microfluidic simulation, the chip design as well as the microfabrication. J.W.v.T. suggested to combine microfluidics with light sheet microscopy and performed the whole-brain in vivo imaging and image analysis. P.E. developed the glass surface smoothening. The manuscript was jointly written by K.M., J.W.v.T., R.W.K., and A.D.

## Competing interests

The authors declare no competing interests.
