## [Peer Review File · Communications Biology]

Reviewers' comments:

Reviewer #1 (Remarks to the Author):

The manuscript "NeuroExaminer: an all-glass microfluidic device for whole brain in vivo imaging" by Dr. Mattern et al is describing all-glass device (NeuroExaminer) that is compatible with whole brain in vivo imaging using light-sheet microscopy. Impressive data has been presented in the manuscript and in the supplementary.

I have only minor criticisms:

At the page for printed:

... The values from the first 5 frames (21-25) were averaged and defined as F_0 , and calcium activity was then calculated for the frames 26-50 as $(F - F_0)/F_0 = \Delta F/F_0$ (Figure 2G-G''').

It seems it is a sort of "first frame analysis" which is relatively common in the imaging studies. If so, it requires reference.

Anyway, could authors explain the reason to use first 5 frames, not just one frame? I guess to reduce the noise? If so, please add a couple of sentences explaining signal-to-noise ratio improving.

Besides that, the data is original, the manuscript is organized well, and written clearly. I will be happy to recommend the manuscript for the publication after minor corrections, suggested above.

Reviewer #2 (Remarks to the Author):

In their manuscript entitled "NeuroExaminer: an all-glass microfluidic device for whole-brain in vivo imaging", Mattern et al. introduce a microfluidic device that can mount and immobilize zebrafish larvae for subsequent light-sheet imaging under presentation of chemical stimuli. The chambers are produced using laser-machining of borosilicate glass which offers better optical quality than microfluidic devices manufactured from PDMS. After an introduction of the method (Figure 1) the authors proceed to showcase experimental whole-brain calcium imaging data acquired with the device mounted in a commercial light-sheet microscope (Figure 2).

In the long run, the advantage of the presented device is its scalability to high-throughput parallel imaging of many larva at once which could be of significant interest to drug screening studies. In its present form, another (albeit unstated) advantage is the horizontal mounting of the larva which is a more physiological orientation unlike the vertical mounting approach used in early whole brain imaging studies (Ahrens et al, Nat Meth. 2013). It is thus of interest to the community and I recommend publication after addressing my comments below.

Major points:

1. My main criticism of the manuscript is that the authors extensively state the suitability of the device for delivering chemical stimuli, for example by highlighting their CFD simulations but do not proceed to showing this experimentally - for example by providing Calcium traces over the course of an odorant exposure experiment. They indicate in line 97 that this will be included in future studies but as the broad utility of the device rests on this capability and the whole design was carried out with it in mind, I suggest to include a proof-of-concept demonstration in the manuscript.

2. The authors state that (Line 31-35) that a device made from borosilicate glass is better suited for “high-resolution image acquisition not suffering from undesired refractions and scattering”. Therefore, I suggest to quantify the resolution offered by the microscope in combination with the microfluidic device by providing PSF measurements (xyz profiles) of beads (for example in agarose) placed inside the imaging compartment in the open and closed configuration.

Minor points:

1. The line rendering in Figure 1B is confusing to interpret, I suggest to label the individual parts of the glass chip in a photograph of a single device (similar to 1E) instead.

2. As the Leica SP8 DLS light-sheet geometry is highly confusing for people who have never seen this instrument or heard about it (the light-sheet is excited from below through the inverted stand, bounces off miniature fold mirrors held by a sleeve around the detection lens; the detection path uses a camera in the condenser arm of an inverted microscope), I suggest to extend/improve Figure 1C. For example, it should be pointed out that the fish is horizontal and the detection axis points upwards. It might also help to show a photograph of a mounted device under the microscope to orient readers.

3. If calcium traces are shown for single cells (as stated in the caption of Figure 2G), the location of the cells should be indicated in the overview projections and ideally in small insets at higher magnification.

4. In general, a volume rate of 0.29 vol/s is too low to accurately measure neuronal activity. The showcased single-cell calcium traces therefore have very symmetrical peaks and no clear exponential decay as expected from a calcium indicator. I assume that this is a limitation of the commercial imaging setup (current state-of-the-art experiments run at around 3 Hz whole-brain rate – i.e. Mu et al, Cell, 2019). If the authors intend to showcase calcium traces, it might make sense to acquire fewer planes to reach volume rates >1 Hz.

5. Please indicate the units of the time stamp (min:s in the supplementary videos) – in addition, it does not make sense to include 3 significant digits for the time stamp at such a low volume rate.

6. To strengthen the manuscript, it would be great to show a video of a larva being positioned & fixated in the chamber before imaging.

Reviewer #3 (Remarks to the Author):

The current manuscript presents a microfluidic chip to analyze whole-brain neural activity combining chemical stimuli application with high spatiotemporal precision and imaging of entire zebrafish brains with cellular resolution utilizing light-sheet microscopy. To achieve high resolution imaging, the microfluidic chip was fabricated out of glass. The design is very successful and the community will benefit from this device, definitely. However, the journal is on biology, and I do not see any novelty on the biology side. Since only the monitoring of brain activity is presented, nothing novel at this point on the biology side.

The design of the chip is novel and the fabrication is challenging. However, although the design of the chip is novel, the scientific novelty on the design part is again marginal. It is a just a brilliant idea with routine engineering effort. I believe the authors are planing some novel measurements via NeuroExaminer for future studies, but I do not think in this form the paper is suitable to be published

in this journal. I recommend authors to include presentation of some novel biological aspect before publication in this journal (reporting of effect of a certain chemical stimuli which has not been discussed in the literature and/or which cannot be detect with conventional methods). Otherwise, considering the design and fabrication aspects, the study is more suitable for an engineering journal.

Here are my additional comments:

1. The figures in Figure 1 is not clear, especially CAD figures (1B and 1C). Since there are many lines on the figures, it is very hard to distinguish different layers. A blow-up figure may be very useful, which shows individual layers.
2. No details are given about the simulations? Most importantly, what exactly learnt from the simulations? Are simulation 2D or 3D? At least some details should be presented in the appendix.
3. To clarify the fabrication process, a schematics of the step by step fabrication would be very informative (at least at the appendix).
4. Figure 1E is also very hard to understand. A blow-up figure together with step-by-step fabrication figure may be beneficial also to understand Figure 1E.

Response to the reviewer comments

Reviewer #1: *The values from the first 5 frames (21-25) were averaged and defined as F_0 , and calcium activity was then calculated for the frames 26-50 as $(F - F_0)/F_0 = \Delta F/F_0$ (Figure 2G-G'''). It seems it is a sort of "first frame analysis" which is relatively common in the imaging studies. If so, it requires reference. Anyway, could authors explain the reason to use first 5 frames, not just one frame? I guess to reduce the noise? If so, please add a couple of sentences explaining signal-to-noise ratio improving.*

Response of the authors: Although zebrafish larvae are generally tightly locked in the "fish fixator" of the NeuroExaminer, small movements may nevertheless occasionally occur (it is expected that larval movements will be further reduced, once there is a continuous flow through the media inlet of the device simulating a more natural environment). Small movements of the larvae, in turn, may result in image shifts which require image registration before data analysis. The frames 26-50 were chosen for data analysis of the depicted larva, since no movement could be detected during this time interval. Consequently, we chose 5 frames (21-25) just before single-cell calcium analysis in order to measure neuronal activity during this time interval most accurately¹. The reason why we chose 5 frames and not a single frame, e.g. frame 25, is due to the fact that averaging the calcium signal across 5 frames provides a much more robust and accurate measure and also a higher signal-to-noise ratio^{1,2}. For example, if the analyzed region of interest (ROI) shows a very bright fluorescence signal in frame 25, similarly bright neuronal activity during frames 26-50 may not be detected, or, on the other hand, if the analyzed ROI shows a very dim signal, even slight increases in fluorescence intensity would result in disproportionately large values of $\Delta F/F$.

Reviewer #2: *In the long run, the advantage of the presented device is its scalability to high-throughput parallel imaging of many larva at once which could be of significant interest to drug screening studies. In its present form, another (albeit unstated) advantage is the horizontal mounting of the larva which is a more physiological orientation unlike the vertical mounting approach used in early whole brain imaging studies (Ahrens et al, Nat Meth. 2013).*

Response of the authors: The authors thank the reviewer for pointing this out. We have now explicitly stated the horizontal mounting of the larva in the second paragraph when describing the NeuroExaminer design and also included a reference to Ahrens et al, *Nat Meth.* 2013 here. Furthermore, we provide an outlook on the potential of our microfluidic device in drug screening studies and high-throughput imaging in the last paragraph.

Reviewer #2: *The authors extensively state the suitability of the device for delivering chemical stimuli, for example by highlighting their CFD simulations but do not proceed to showing this experimentally - for example by providing Calcium traces over the course of an odorant exposure experiment. They indicate in line 97 that this will be included in future studies but as the broad utility of the device rests on this capability and the whole design was carried out with it in mind, I suggest to include a proof-of-concept demonstration in the manuscript.*

Response of the authors: The reviewer is right: the NeuroExaminer was built to eventually apply chemical stimuli with a high spatiotemporal precision and to monitor the resulting neuronal response throughout the whole-brain *in vivo*. Yet, when starting this project, it was not clear whether a light-sheet imaging approach of brain tissue could be achieved in chemically inert glass chambers and neither a suitable chamber design nor a practical fabrication procedure existed. We do provide data of successful Ca^{2+} -activity recording with cellular resolution in resting larvae that apparently tolerate confinement inside the designed glass chamber without physiological damage. Indeed, liquid stimuli application will be the major focus of future investigations, as we are currently building a setup that will allow us to generate this data. Here, we provide computational fluid dynamics (CFD) simulations of stimuli generation and propagation, however, which are generally very reliable in low Reynolds Number regimes indicating laminar flow as it

is typical in microdevices. The knowledge gain achieved by CFD simulations is now described in more detail and more clearly in the revised version of the manuscript.

We estimate that we will require approximately 6 – 12 months until we can provide data with stimuli application, which will then subsequently have to be followed by extensive data processing and analysis. This will be well beyond the 3 months period that is generally provided for a revised manuscript in Communications Biology and the amount of expected data will exceed the format of a short communication. We are convinced that our advance over the past years in combining chamber design and manufacturing with light sheet imaging provides intriguing information and valuable knowledge for the growing discipline of microfluidic device development for physiologically interrogating cells, organs, organoids, and organisms. Therefore, we believe that the availability of these data should not be delayed further, but made accessible for adapting glass-based chamber development to a wide range of biological contexts. We hope the reviewer agrees with the view that a short communication demonstrating a technological advance for biological data collection is within the scope of Communications Biology and will be of interest to a broad readership.

Reviewer #2: I suggest to quantify the resolution offered by the microscope in combination with the microfluidic device by providing PSF measurements (xyz profiles) of beads (for example in agarose) placed inside the imaging compartment in the open and closed configuration.

Response of the authors: We used 200 nm fluorescent microspheres (Dragon Green (FSDG002; 480 nm (excitation maxima), 520 nm (emission maxima), Bangs Laboratories, Fishers, IN, USA) in agarose (diluted 1:10⁶ from the originally supplied solution) and the methodology described in Cole et al *Nature Protocols* 6, 1929-1941 (2011) to determine the point spread function (PSF) in the open and closed configuration of the NeuroExaminer (Figure 1). In order to visualize the beads, we had to increase the laser power and use a longer exposure time (100 ms versus 20 ms) than what we normally use for calcium imaging; all other values (2x2 binning and “extended” light sheet) were kept identical. When imaged, the beads appeared with an elliptical shape in the xy plane in both the open and the closed system.

Figure 1 Full width at half maximum (FWHM) values for x, y, and z of the point spread function (PSF) in the open (left; n=10 beads) and closed (right; n=12 beads) configuration of the NeuroExaminer. Horizontal black and vertical lines depict the mean and standard deviation, respectively. Theoretically calculated values are depicted with a thin grey rectangle at 1.1 µm for x and y, and 5.9 µm for z.

In the open system (n=10 beads), we obtained full width at half maximum (FWHM) values of 1.20 ± 0.11 , 1.40 ± 0.17 , and 7.80 ± 0.31 for x, y, and z, respectively (Figure 1, left). In the closed system (n=12 beads), we obtained FWHM values of 1.76 ± 0.48 , 1.69 ± 0.34 , and 10.63 ± 3.02 for x, y, and z, respectively (Figure 1, right). Whereas the FWHM values for x and y are similar in the open and the closed configuration of the microfluidic device, the increase in z in both the mean and the standard variation in the closed configuration appears to corroborate our previous findings during calcium imaging that optical

planes in deeper brain layers (more ventral) are generally less well resolved than those in more superficial layers (more dorsal) in the closed compared to the open system. The experimentally obtained values are within the range of the theoretically calculated for x and y (theoretical value: $1.1 \mu\text{m}$) and slightly higher for z (theoretical value: $5.9 \mu\text{m}$).

Reviewer #2: 1. *The line rendering in Figure 1B is confusing to interpret, I suggest to label the individual parts of the glass chip in a photograph of a single device (similar to 1E) instead.*

Response of the authors: We have replaced Figure 1B with a schematic 3D overview of the microfluidic chip. In this new figure that also provides a better perspective of the chip, the individual parts are color-coded and labelled. In addition, we now also depict a photograph of a single microfluidic glass chip in Figure 1E. Together this should provide the reader with a much better understanding of the chip design.

Reviewer #2: *As the Leica SP8 DLS light-sheet geometry is highly confusing for people who have never seen this instrument or heard about it (the light-sheet is excited from below through the inverted stand, bounces off miniature fold mirrors held by a sleeve around the detection lens; the detection path uses a camera in the condenser arm of an inverted microscope), I suggest to extend/improve Figure 1C. For example, it should be pointed out that the fish is horizontal and the detection axis points upwards. It might also help to show a photograph of a mounted device under the microscope to orient readers.*

Response of the authors: We have made significant changes to Figure 1C to better orient the reader of our particular imaging setup. We have enlarged the perspective, provided contrast and color to the detection objective, extended the light sheet's beam path towards the illumination objective (not depicted), and for a clearer presentation excluded the detailed channel structure from the microfluidic chip. In addition, we have also provided a new supplementary Figure 1, in which we provide a detailed view (both schematic and through photographs) of the main components of our imaging setup, including the imaging chamber and the microfluidic chip together with the detection objective with its two mirrors. We hope that this facilitates the general understanding of our setup and is in particular useful to people not familiar with the particularities of Leica's digital light sheet microscope.

Reviewer #2: *If calcium traces are shown for single cells (as stated in the caption of Figure 2G), the location of the cells should be indicated in the overview projections and ideally in small insets at higher magnification.*

Response of the authors: We would like to thank the reviewer for this suggestion and have included an additional time-lapse movie (supplementary movie 7) of a single z -plane ($z=8, 70 \mu\text{m}$) that depicts the analyzed cells location and that was used to generate the calcium traces shown in Figure 2G. The cells are color-coded with respect to their location in the telencephalon (blue), habenula (green), optic tectum (magenta), cerebellum (orange), and hindbrain (yellow). We have also included small insets at higher magnification depicting the calcium transients in the analyzed cells.

Reviewer #2: *In general, a volume rate of 0.29 vol/s is too low to accurately measure neuronal activity. The showcased single-cell calcium traces therefore have very symmetrical peaks and no clear exponential decay as expected from a calcium indicator. I assume that this is a limitation of the commercial imaging setup (current state-of-the-art experiments run at around 3 Hz whole-brain rate – i.e. Mu et al, Cell, 2019). If the authors intend to showcase calcium traces, it might make sense to acquire fewer planes to reach volume rates $>1 \text{ Hz}$.*

Response of the authors: As the reviewer suspected, the commercial Leica DLS microscope controlled through the LAS X software is limited in its capability for fast volumetric imaging. With this setup, imaging *Tg(elavl3:H2B-GCaMP6s)* larvae in the *crystal* background requires an exposure time of approximately 20 ms in our experience. With this exposure time, acquiring 21 z -planes spaced $10 \mu\text{m}$ apart (i.e. a z -

depth of 200 μm that is comparable to Ahrens et al, Nat Meth 2013³ although it contains nearly only half as many z-planes in total (21 versus 41)) results in volume rate of 0.29 vol/s. On the Leica DLS microscope, volumes rates of ~ 1 Hz of calcium-imaging can only be achieved if the number of z-sections is limited to around 5. A volume of 40 μm though is sufficient for most Ca^{2+} -imaging studies addressing neuronal activity in distinct brain compartments in zebrafish larvae thereby providing adequate spatial and temporal resolution for determining the physiological response of neuronal populations upon stimulated neuronal network activity. To demonstrate the increased temporal resolution in a 40 μm volume of brain tissue we have included an additional movie (supplementary movie 9) in which we imaged 5 z-planes each 10 μm in the “closed” NeuroExaminer at 1,1 Hz (with an exposure time of 17 ms). We have also included another figure (supplementary Figure 2) to detail the results of calcium traces acquired from single cells in the optic tectum, cerebellum, and hindbrain.

Reviewer #2: *Please indicate the units of the time stamp (min:s in the supplementary videos) – in addition, it does not make sense to include 3 significant digits for the time stamp at such a low volume rate.*

Response of the authors: The units of the time stamps in supplementary movies 3-6 are minutes:seconds:milliseconds. Two digits are provided for minutes and seconds. In the newly created supplementary videos 7-10 for the revised version of our manuscript, the time units are now specified directly within the movies (mostly in seconds [s]; with the exception of movie 9, in which the time is specified in min:sec).

Reviewer #2: *To strengthen the manuscript, it would be great to show a video of a larva being positioned & fixated in the chamber before imaging.*

Response of the authors: We have provided an additional video (supplementary movie 8) that shows a 6 dpf old larva positioned in the “closed” NeuroExaminer after being inserted through the fish sluice (outlined in cyan in supplementary movie 8; related to Figure 2D) and just before imaging as viewed through the oculars of the Leica DLS microscope (outlined in magenta in supplementary movie 8). We thank the reviewer for this suggestion, as we agree that this video together with supplementary Figure 1A-C' will help the reader to better understand the positioning of the analyzed larvae within the chamber and the particular Leica DLS microscope setup in general.

Reviewer #3: *The design is very successful and the community will benefit from this device, definitely. However, the journal is on biology, and I do not see any novelty on the biology side. Since only the monitoring of brain activity is presented, nothing novel at this point on the biology side.*

Response of the authors: As this point of criticism is somewhat redundant to the next point of concern, we will address these two concerns about limited biological novelty jointly in our answer below.

Reviewer #3: *The design of the chip is novel and the fabrication is challenging. However, although the design of the chip is novel, the scientific novelty on the design part is again marginal. It is a just a brilliant idea with routine engineering effort. I believe the authors are planing some novel measurements via NeuroExaminer for future studies, but I do not think in this form the paper is suitable to be published in this journal. I recommend authors to include presentation of some novel biological aspect before publication in this journal (reporting of effect of a certain chemical stimuli which has not been discussed in the literature and/or which cannot be detect with conventional methods).*

Response of the authors: We very much appreciate the reviewer's point of view to consider combining light sheet microscopy with microfluidics to investigate neural circuits throughout the whole brain *in vivo* a “brilliant idea”. We also agree with the reviewer that the novelty of our presented data with respect to neurobiological mechanistic insights is limited. Yet, as explained in detail in our response to reviewer 2 above, we are convinced that advancements in biological *in vivo* data acquisition in interdisciplinary fields such as microfluidic biosensor development are of significant relevance to the biological community too.

Although biologists may possibly not be able to fully appreciate the engineering steps it takes to design, fabricate and optimize miniature microfluidic glass chambers, we believe to serve a growing community in the field of *in vivo* metrology by providing the design strategy, manufacturing routines and simulation approaches using all-glass micro-instrumentation. Clearly, since biologists are largely the end users of such devices the readership that should be targeted are mainly biologists and not engineers. We do hope that readers of Communications Biology are excited about such interdisciplinary strategies and approaches though.

Reviewer #3: *The figures in Figure 1 is not clear, especially CAD figures (1B and 1C). Since there are many lines on the figures, it is very hard to distinguish different layers. A blow-up figure may be very useful, which shows individual layers.*

Response of the authors: We have revised Figure 1B, C and added an additional picture of a single microfluidic chip in E. Please see also our answer above to reviewer #2 who had raised similar points. Furthermore, we now also provide an additional figure (supplementary Figure 1) that provides a schematic overview but also detailed photographs of our imaging setup.

Reviewer #3: *No details are given about the simulations? Most importantly, what exactly learnt from the simulations? Are simulation 2D or 3D? At least some details should be presented in the appendix.*

Response of the authors: Further details about the simulations are now given in the revised manuscript including flow rates and Reynolds number. We also point out which conclusions we could draw from the simulations, in particular how fast and how precisely located the stimuli can be applied to the larva head using the closed and also a partially opened variant of the NeuroExaminer. The results of the simulations are represented in revised form in Figure 1A'-A". The simulations are performed in 3D as shown by the new streamline pattern images in Figure 1A". The displayed 2D concentration distributions in Figure 1A' correspond to the middle plane of the device and the surface of the fish larvae, as explained in the figure caption. Two new supplementary movies (supplementary movies 1-2) illustrating the arrival of the stimuli at the larva head with the resulting time dependent concentration distribution were also created. As pointed out in the main text, the Reynolds number is low and simulations can be considered as very reliable.

Reviewer #3: *To clarify the fabrication process, a schematics of the step by step fabrication would be very informative (at least at the appendix).*

Response of the authors: A representation of the fabrication process can now be found in supplementary Figure 1D.

Reviewer #3: *Figure 1E is also very hard to understand. A blow-up figure together with step-by-step fabrication figure may be beneficial also to understand Figure 1E.*

Response of the authors: As mentioned above, we have largely revised Figure 1. A single microfluidic chip is now newly depicted in Figure 1E (thus Figure 1E became Figure 1F) and an overview of the step-by-step fabrication process is now shown in supplementary Figure 1D.

References

1. Chen, T.-W. *et al.* Ultrasensitive fluorescent proteins for imaging neuronal activity. *Nature* **499**, 295–300 (2013).
2. Panier, T. *et al.* Fast functional imaging of multiple brain regions in intact zebrafish larvae using selective plane illumination microscopy. *Front Neural Circuits* **7**, 65 (2013).
3. Ahrens, M. B., Orger, M. B., Robson, D. N., Li, J. M. & Keller, P. J. Whole-brain functional imaging at cellular resolution using light-sheet microscopy. *Nat Meth* **10**, 413–420 (2013).

REVIEWERS' COMMENTS:

Reviewer #2 (Remarks to the Author):

The authors have enhanced their manuscript and clarified several of the figures and supplementary videos. However, the central criticism I mentioned - the lack of experimental verification and validation of the stimulus delivery option was not addressed. While it is clear that this requires a significant amount of time, it would also massively strengthen the manuscript and potentially lead to much more interest in the biological community and thus make the manuscript much more citable. In addition, I dislike that the authors phrased several sentences of their manuscript in a way that appears to be misleading, for example in the abstract:

" Both techniques, however, have not been combined to analyze whole-brain neural activity while simulating sensory environments." - implicitly suggesting that they are combined here.

In addition:

"Both NeuroExaminer variants enable a precise spatial and temporal control (with subsecond resolution) of exposure to varying stimuli." - which is only simulated, not supported experimentally in any way.

As the ongoing COVID-19 pandemic probably prohibits the authors from experimental work, I suggest to either significantly extend the reviewing period to allow these experiments to happen or to transfer the manuscript to a more engineering-centric journal.

To summarize: In its current form, it is an acceptable engineering article, however, it lacks significant biological novelty to be suitable for a biology-centric journal.

Reviewer #3 (Remarks to the Author):

The authors have addressed most of the comments quite satisfactorily. However, my comment on the novelty of the study is still on the table. A similar concern was also mentioned by Reviewer#2. Considering the manuscript is a short communication, inclusion of the discussion about further studies, and significant relevance of the current technology for the biological community, I feel comfortable about the publication of the present manuscript on this journal.

Response to the reviewer comments

Reviewer #2: *The authors have enhanced their manuscript and clarified several of the figures and supplementary videos. However, the central criticism I mentioned - the lack of experimental verification and validation of the stimulus delivery option was not addressed. While it is clear that this requires a significant amount of time, it would also massively strengthen the manuscript and potentially lead to much more interest in the biological community and thus make the manuscript much more citable. In addition, I dislike that the authors phrased several sentences of their manuscript in a way that appears to be misleading, for example in the abstract:*

"Both techniques, however, have not been combined to analyze whole-brain neural activity while simulating sensory environments." - implicitly suggesting that they are combined here.

Response of the authors: We agree with Reviewer#2 that we are currently not able to monitor neural activity while simulating sensory environments, since we have not implemented a stimulus application yet. We have thus deleted the last part of the sentence and it now reads: "Both techniques, however, have not been combined to monitor whole-brain neural activity yet."

Combining microfluidics with light sheet microscopy was the main idea and driving force behind our study, and although we have currently not fully accomplished this yet, we have made progress towards this goal. Therefore, the authors believe it is important to state the main idea of the project in the abstract, but at the same time we also agree with Reviewer#2 that there is still work ahead, namely stimuli application.

Reviewer #2: *In addition:*

"Both NeuroExaminer variants enable a precise spatial and temporal control (with subsecond resolution) of exposure to varying stimuli." - which is only simulated, not supported experimentally in any way.

Response of the authors: We have modified this sentence and it now reads: "In the simulations, both NeuroExaminer variants enable a precise spatial and temporal control (with subsecond resolution) of exposure to varying stimuli." thereby making it clear that this data is currently only simulated.

Reviewer #3: *The authors have addressed most of the comments quite satisfactorily. However, my comment on the novelty of the study is still on the table. A similar concern was also mentioned by Reviewer#2. Considering the manuscript is a short communication, inclusion of the discussion about further studies, and significant relevance of the current technology for the biological community, I feel comfortable about the publication of the present manuscript on this journal.*

Response of the authors: We have extended the discussion to make it clear that the NeuroExaminer may be useful to a variety of light sheet setups (commercial and non-commercial) and, therefore, many biological laboratories could possibly make use of the developed technology and apply it to investigate their own biological questions. Moreover, since zebrafish larvae can be imaged repeatedly and the setup is in principle scalable, neuromodulatory screening studies that directly analyze neuronal activity rather than behavior, for example, may also become feasible with the developed microfluidic chip in zebrafish soon.